# Postharvest Application of Potato Starch Edible Coatings with Sodium Benzoate to Reduce Sour Rot and Preserve Mandarin Fruit Quality

**Lourdes Soto-Muñoz** [1,2], **María B. Pérez-Gago** [1], **Victoria Martínez-Blay** [1] **and Lluís Palou** [1,*]

[1]   Centre de Tecnologia Postcollita (CTP), Institut Valencià d'Investigacions Agràries (IVIA), Montcada, 46113 Valencia, Spain
[2]   Facultad de Química, Universidad Autónoma de Querétaro, Centro Universitario S/N, Santiago de Querétaro 76010, Mexico
*   Correspondence: palou_llu@gva.es; Tel.: +34-963-424-117

**Abstract:** Starch is a biodegradable polymeric carbohydrate that can easily form films and coatings and can readily be obtained from some food industry by-products and wastes, which may contribute to the circular bioeconomy. In this work, we studied the potential of two edible coating emulsions based on pregelatinized potato starch (PPS) and glyceryl monostearate (GMS) alone (F6 and F10) or formulated with the food additive sodium benzoate (SB, 2%) (F6/SB and F10/SB) to control sour rot, an important citrus postharvest disease caused by the fungus *Geotrichum citri-aurantii*, and maintain postharvest quality of cold-stored 'Orri' mandarins. The PPS-GMS coating application was compared to dipping in water (uncoated controls) and dipping in a 2% SB (*w/v*) aqueous solution. The results showed that the coating F10/SB was the most promising treatment to control sour rot on mandarins, with reductions in disease incidence with respect to the uncoated control samples of 94, 69, and 55% after 2, 4, and 6 weeks of storage at 5 °C, respectively. Coatings formulated without SB were ineffective. Regarding fruit quality, the coating F10 was the most effective to reduce weight loss, maintain firmness, and provide gloss on mandarins stored at 5 °C for up to 6 weeks followed by a shelf-life period of 1 week at 20 °C. The addition of SB to the PPS-GMS coatings adversely affected these coating properties, but the coating F10/SB still reduced weight loss compared to uncoated controls without negatively affecting the fruit physicochemical (juice titratable acidity, soluble solids content, and volatiles content) and sensory quality (overall flavor, off-flavors, external aspect). Overall, the coating F10/SB showed the greatest potential for commercial use as an efficient and environmentally friendly alternative to conventional fungicides and waxes for sour rot control and quality preservation of cold-stored mandarins.

**Keywords:** *Citrus reticulata*; postharvest quality; antifungal starch coatings; *Geotrichum citri-aurantii*





## 1. Introduction

Citrus sour rot, caused by the pathogenic fungus *Geotrichum citri-aurantii*, is an important postharvest disease that has become a major concern in many citrus producing and exporting countries worldwide [1]. The incidence and consequent economic importance of sour rot has increased in recent years, especially in citrus regions with a Mediterranean-type climate, due to the increment of extreme weather events, such as violent rainfalls with large drops, hail, and gusts of intense wind that cause splashes, blows, and rind wounds in the fruits [2,3]. Inoculum of *G. citri-aurantii* is present in soil particles in citrus orchards and reaches the fruits, particularly those in the lower part of the tree, via water splash during the mentioned weather episodes, especially heavy rains. Once located on the surface of mature fruits, arthrospores of the fungus can invade the rind of the fruit through injuries and bruises, especially deep wounds that extend into the albedo. Successful infections on immature fruits or through shallow rind injuries are more difficult to occur [1,2]. Sour rot

lesions on citrus fruits are watery, soft, and of glutinous nature, and under high relative humidity (RH) a yeasty, wrinkled layer of whitish mycelium is formed on their surface. Infected tissues have a characteristic yeasty, vinegary odor that attracts fruit flies and other insects that may also contribute to disease dispersion [1].

Treatments with conventional chemical fungicides, such as imazalil, fludioxonil, pyrimethanil, o-phenyl phenol, and sodium o-phenyl phenate, serve as the primary means to commercially control postharvest diseases of citrus fruits [1,4]. However, these agrochemicals are devoted to the control of Penicillium molds, which are generally the most important citrus postharvest diseases causing economical losses, and they do not control sour rot effectively [5]. Guazatine and propiconazole are fungicides with proven, high effectiveness against *G. citri-aurantii*, but they have been withdrawn from the market in the European Union (EU) because of their high toxicity to humans and the environment. Therefore, the lack of authorized fungicidal active ingredients for sour rot control in the EU and other citrus producing areas is a serious threat for the marketing of high-quality citrus fruits, and there is an increasing need for safe and effective nonpolluting alternative strategies to control citrus postharvest sour rot [6,7].

The use of some common salts classified as generally recognized as safe (GRAS) compounds or food additives is a promising eco-friendly alternative to synthetic fungicides for citrus sour rot management [6,8]. Recently, a study conducted by our group showed that dipping fruit in aqueous solutions of sodium benzoate (SB) at 3% ($w/v$) for 60 s at 20 or 50 °C has a strong curative effect, similar to that of the chemical fungicide propiconazole, against sour rot on mandarins and oranges artificially inoculated with *G. citri-aurantii* [7]. In addition, Smilanick et al. [9] reported that immersion of inoculated lemons in 1% ($w/v$) sodium bicarbonate solutions for 1 min at 24 °C reduced the number of viable spores of the pathogen. Sour rot incidence was also reduced moderately after treatment with a potassium sorbate solution at 1% ($w/v$) for 30 s at 50 °C on lemons inoculated with *G. citri-aurantii* 24 h before treatment [10]. Furthermore, sodium salicylate, boric acid, EDTA, sodium dehydroacetate, and sodium silicate also exhibited significant potential for sour rot control on mandarin fruit [11,12]. However, some of these aqueous treatments lacked persistence and/or adversely affected the quality attributes of treated fruit.

Postharvest use of composite edible coatings (ECs) based on food grade, biodegradable materials is a promising approach to maintain postharvest quality and extend the life of fresh horticultural produce, particularly fresh fruits and perishable fruit vegetables [13,14]. Starch-based coating matrixes can be used for this purpose due to their compatibility with a wide range of functional compounds, relative low cost, and desirable film characteristics. Furthermore, starch is a biodegradable polymeric carbohydrate that can be easily obtained from different sources, including food industry by-products and wastes, which can contribute to the circular bioeconomy. In general, starch films are odorless, transparent, taste-free, non-toxic, and highly impervious to $O_2$. However, typically, starch films have high water vapor permeability and additional ingredients are needed in the emulsion to obtain coating matrixes suitable to reduce water loss, and consequently, weight loss, during long-term storage of horticultural produce. Hence, the addition of hydrophobic compounds such as lipids and other minor ingredients such as plasticizers and emulsifiers is reported to improve the properties of starch-based films [15]. Thus, composite starch matrixes formulated with appropriate lipidic materials and other adjuvants have successfully modified the gas composition within the fruit and reduced weight loss through the regulation of gaseous exchange ($O_2$, $CO_2$, and water vapor) between coated fruits and vegetables and the ambient atmosphere [16]. An important advantage of ECs is the possibility of incorporating food additives or GRAS salts as additional ingredients into the basic coating formulation [14]. Nevertheless, the addition of these additional functional ingredients to coating matrixes can lead to the formation of unstable or incompatible emulsions showing phase separation, modify the functional properties of the coating, and/or change the organoleptic profile of the coated product. In some cases, the additional ingredient can even lose their functionality when reacting with some original coating components [13]. There is, therefore,

a general need to develop tailored EC emulsions for each particular purpose and fresh commodity [14,17].

Available information in the literature on starch-based films formulated with antimicrobial agents is limited. Marín et al. [18] found that corn starch-based coatings applied as carriers of the antagonistic yeast *Candida sake* reduced the incidence of postharvest disease caused by *Botrytis cinerea* on coated grapes. Likewise, potato starch coatings containing *Lactobacillus plantae* also reduced grape gray mold. The addition of natamycin/methyl-β-cyclodextrin complex to corn starch ECs reduced weight loss, delayed ripening, and inhibited gray mold on cherry tomato fruits during storage at 22 °C [19]. Moreover, different thyme essential oils amended to starch–gellan films showed high antifungal activity in vitro and, furthermore, significantly reduced gray mold caused by *B. cinerea* and black spot caused by *Alternaria alternata* on persimmons and apples during incubation at 25 °C of artificially inoculated and coated fruits [15]. In another research work, the GRAS salt potassium sorbate was better retained when it was incorporated into a starch-based EC formulation, which enhanced its antifungal activity and persistence against spoilage molds on refrigerated apples, tomatoes, and cucumbers [20].

In a previous study, Soto-Muñoz et al. [21] developed and optimized a formulation for an antifungal composite EC based on pregelatinized potato starch and glyceryl monostearate (PPS-GMS) with the addition of SB as antifungal ingredient. This formulation presented curative activity against postharvest diseases caused by *Penicillium digitatum*, *Penicillium italicum*, or *G. citri-aurantii* on artificially inoculated 'Orri' mandarins coated and incubated at 20 °C for 7 days. In addition, this formulation also reduced mandarin weight loss without negatively affecting the overall fruit quality and consumer acceptability. Nevertheless, no information is available about the performance of this type of coatings during commercial long-term cold storage of mandarins, and such data are necessary to validate the performance of ECs under practical commercial conditions. Therefore, the objectives of this research work were to: (i) assess the ability of antifungal PPS-GMS ECs formulated without or with SB to reduce the development of sour rot and (ii) determine the effects of EC application on the physicochemical and sensorial quality of 'Orri' mandarins during long-term cold storage at 5 °C plus shelf life at 20 °C.

## 2. Materials and Methods

### 2.1. Mandarin Fruits

This research was conducted with 'Orri' hybrid mandarins (*Citrus reticulata* Blanco) collected in commercial orchards in the Valencia citrus growing area (Spain). Fruits were harvested at commercial maturity and used the same or the following day. Mandarins were manually selected to discard diseased and injured fruits. Selected fruits were randomized and surface disinfected before fungal inoculation or postharvest treatment application. For disinfection, fruits were immersed in a sodium hypochlorite solution (0.5%, *v/v*) for 4 min, rinsed with tap water, and allowed to air-dry at ambient temperature.

### 2.2. Preparation of Edible Coating Formulations

The main composite coating matrix of the ECs used in the present research was prepared with optimized amounts of PPS (Quimidroga, S.A., Barcelona, Catalonia, Spain), GMS (Italmatch Chemicals Spa, Barcelona, Catalonia, Spain), glycerol (Panreac-Química S.A., Barcelona, Catalonia, Spain), and sunflower lecithin and diacetyl tartaric acid esters of mono-diglycerides (TATEM) (Lasenor S.A., Barcelona, Catalonia, Spain) as emulsifiers. These two emulsifiers were used at a ratio of 1:1, dry basis (d.b.), while the ratio GMS:emulsifier was 2:1 (d.b.). In a previous study, we determined the optimized amount of these coating components by applying Box–Behnken response surface methodology [21]. The GRAS salt SB (Sigma-Aldrich Química S.A., Tres Cantos, Madrid, Spain) at 2% (*w/w*) was incorporated into the main matrixes as an additional antifungal ingredient. The mentioned ingredients were combined in different proportions to formulate the assayed emulsions: two coatings named F10 and F6 were formulated without SB and two other

coatings named F10/SB and F6/SB were the emulsions F10 and F6 including SB at 2% (Table 1). The methodology described by Valencia-Chamorro et al. [17] was followed for the preparation of the emulsions. Coating emulsions were kept overnight at 5 °C before use in the experiments.

**Table 1.** Composition on a dry basis (d.b., %) of the experimental edible coatings used in this study.

| Components | F10 | F6 | F10/SB | F6/SB |
|---|---|---|---|---|
| PPS | 48.3 | 28.6 | 32.0 | 18.2 |
| GMS | 13.2 | 28.6 | 8.8 | 18.2 |
| Glycerol | 25.2 | 14.3 | 16.7 | 9.1 |
| Sunflower lecithin | 6.6 | 14.3 | 4.4 | 9.1 |
| TATEM | 6.6 | 14.3 | 4.4 | 9.1 |
| Sodium benzoate (SB; %, w.b.) | - | - | 2.0 | 2.0 |
| SC (%) | 3.9 | 3.5 | 5.9 | 5.5 |

PPS: pregelatinized potato starch; GMS: glyceryl monostearate; TATEM: diacetyl tartaric acid esters of mono-diglycerides; SC: solids content; w.b.: wet basis.

### 2.3. Effect of Coating Application on Mandarin Sour Rot

#### 2.3.1. Inoculum of *G. citri-aurantii*

The isolate of the pathogen *G. citri-aurantii* used in this research (strain NAV-1) was obtained from a decayed orange collected in a fruit packinghouse located in the citrus growing area of Valencia (Spain). The strain was assigned the accession number CECT 13166 in the Spanish Type Culture Collection (CECT, University of Valencia, Valencia, Spain) and it is available in the culture collection of postharvest pathogens of the IVIA Postharvest Technology Center. Arthrospores from 7- to 14-day-old potato dextrose agar (PDA) fungal cultures were transferred in sterile conditions to an aqueous solution containing Tween® 80 (0.05%, Panreac-Química S.A., Barcelona, Catalonia, Spain), filtered through two layers of cheesecloth and adjusted to $1 \times 10^7$ arthrospores/mL by counting the spores with a hemacytometer. Then, the following ingredients were added to enhance the aggressiveness of the arthrospore suspension and facilitate the infection process in citrus fruits: fresh mandarin juice (10%); the chemical fungicide thiabendazole (50 mg/L; Textar® 60 T, Decco Ibérica PostCosecha, S.A.U., Paterna, Valencia, Spain) to avoid the development of green/blue molds in the inoculated wounds from mixed infections by ambient spores of *Penicillium* spp.; and the protein synthesis inhibitor cycloheximide (5 mg/L; Carl Roth GmbH + Co. KG, Karlsruhe, Germany) to avoid rind wound healing.

#### 2.3.2. Fungal Inoculation, Coating Application, and Assessment of Sour Rot

Fungal inoculation of mandarins was performed by immersing a stainless-steel rod with a probe tip 1 mm wide and 2 mm in length into the *G. citri-aurantii* arthrospore suspension and wounding each fruit once in the equator. Inoculated mandarins were incubated at 28 °C and 90% RH for 24 h before treatment application. Treatments applied were: (i) immersion in tap water at 20 °C for 15 s, as uncoated negative control; (ii) coating F10, fruits were coated by immersion for 15 s in the emulsion; (iii) coating F6; (iv) immersion in a 2% SB aqueous solution (*w/v*) at 20 °C for 15 s, as positive control; (v) coating F10/SB; and (vi) coating F6/SB. The emulsions were prepared as described in the previous Section 2.2. Coated fruits were drained by placing them in metallic grids and allowed to air-dry at ambient temperature. Each treatment was applied to 4 replicates of 10 fruits each. Fruits were placed on plastic trays and cold stored at 5 °C and 90% RH to resemble commercial storage practices for Spanish export fruit. The incidence of sour rot was assessed as the percentage of infected fruits and the severity of sour rot as the diameter of the lesion (mm). Both disease incidence and severity were assessed after 2, 4, and 6 weeks of cold storage. The experiment was repeated once.

### 2.4. Effect of Coating Application on Mandarin Quality during Cold Storage

Mandarin external and internal quality was determined at harvest (initial quality) and after 2, 4, and 6 weeks of cold storage at 5 °C followed by 1 week of simulated shelf life at 20 °C. Sixty mandarins per treatment were appropriately selected and randomized and used for fruit quality evaluations. For coating treatments (F6, F10, F6/SB and F10/SB), mandarins were immersed for 15 s in the corresponding coating emulsion, drained, and dried in a heated air-forced tunnel at 45 ± 2 °C for 130 s. Uncoated mandarins were dipped in water at 20 °C for 15 s and used as controls. Another set of fruits was immersed in a 2% SB aqueous solution ($w/v$) at 20 °C for 15 s. The evaluated quality attributes were the following:

#### 2.4.1. Fruit Weight Loss

After treatment, 20 fruits per treatment were individually numbered and weighed with a calibrated analytical balance (Alessandrini® P30, Modena, Italy). The weight measurement of each fruit was repeated at the end of each storage period plus shelf life. Results were expressed as the percentage loss of initial weight as follows: % WL = [(Wi − Wf)/Wi] × (100), where % WL = percentage of weight loss, Wi = initial fruit weight (g), and Wf = final fruit weight (g).

#### 2.4.2. Fruit Firmness

Fruit firmness was evaluated as the percentage of rind deformation, related to initial diameter, using an Instron Universal testing machine (Model 3343, Instron Corp., Canton, MA, USA). Measurements were taken on 20 mandarins per treatment.

#### 2.4.3. Quality of Mandarin Juice

Soluble solids concentration (SSC, °Brix) and titratable acidity (TA, percentage of citric acid) were assessed in 5-mL juice samples (3 replicates of 10 mandarins each per treatment) as described by Valencia-Chamorro et al. [17]. TA was determined with an automatic titrator (Titrator T50, Mettler Toledo, Switzerland) and SSC with a digital refractometer (model ATC-1, Atago® Co., LTD, Tokyo, Japan).

#### 2.4.4. Internal Gas Concentration

The concentration of both $CO_2$ and $O_2$ (kPa) in the internal cavity of 10 mandarins per treatment was measured with a gas chromatograph (GC, Thermo Trace, Thermo Fisher Scientific, Inc., Waltham, MA, USA) according to Valencia-Chamorro et al. [17].

#### 2.4.5. Ethanol and Acetaldehyde Contents

Ethanol (EtC, mg/L) and acetaldehyde (AcC, mg/L) contents were analyzed from the headspace of juice from 3 replicates of 10 mandarins per treatment. Samples were analyzed by gas chromatography following the methodology described by Valencia-Chamorro et al. [17].

#### 2.4.6. Sensory Evaluation

Sensory evaluation of 5 mandarins per treatment was conducted by 15 judges after each cold storage and shelf-life period. Judges were semi-trained volunteers among IVIA personnel. After peeling the rind from the fruit, individual mandarin segments properly identified with 3-digit random codes were offered to the judges. Global acceptance was assessed through a 9-point verbal hedonic scale ranging from 1 to 9, where 1 to 3 = dislike, 4 to 6 = acceptance, and 7–9 = excellent. Panelists also visually ranked the treatments from highest to lowest gloss according to the method proposed by Valencia-Chamorro et al. [17].

### 2.5. Statistical Analysis

For each evaluation date, data from sour rot disease (incidence and severity) on artificially inoculated mandarins were analyzed using a bifactorial analysis of variance (ANOVA) in which the treatment (control, SB solution, and four different coatings) and the

experiment (the trials were repeated once) were the independent factors. Since the factor experiment was not significant, mean values of both experiments are presented. Since sour rot incidence data were proportions derived from counting, they were arcsine-transformed prior to the ANOVA to assure the analysis homoscedasticity. Non-transformed means are presented. Regarding fruit quality evaluations, data on each mandarin quality attribute at each evaluation date were subjected to a monofactorial ANOVA with the treatment as the independent variable. In all cases, Fisher's protected least significant difference test (LSD, $p = 0.05$) was used for separation of means. For sensorial gloss, the differences among treatments were determined using the Friedman test, as it is recommended for ranking by the regulation UNE-ISO 8587 [22]. All statistical analyses were performed with the software Statgraphics Centurion XVIII (18.1.16.0, 2018, Statgraphics Technologies Inc., The Plains, VA, USA).

## 3. Results

### 3.1. Effect of Coating Application on Mandarin Sour Rot

Two PPS-GMS-based composite coatings formulated without (F10 and F6) or with the antifungal ingredient SB at 2% (F10/SB and F6/SB) were evaluated to control disease caused by *G. citri-aurantii* on 'Orri' mandarins artificially inoculated, coated 24 h later, and cold stored at 5 °C for up to 6 weeks. A positive control of 2% SB aqueous solution was also assayed. Surprisingly, Figure 1 shows that regardless of the storage time, both the incidence and severity of sour rot on mandarins coated with F6 and particularly with F10 were higher than those on uncoated control fruit. In contrast, the antifungal ECs F6/SB and F10/SB significantly reduced the incidence of sour rot on artificially inoculated mandarins after 2, 4, and 6 weeks of cold storage and reduced sour rot severity after 4 and 6 weeks. The coating F10/SB was the most effective treatment to control sour rot on coated and cold-stored mandarins, although the differences between the coating F6/SB and the SB aqueous treatment were not always statistically significant. The application of F10/SB produced disease incidence reductions of 94% after 2 weeks, 69% after 4 weeks, and 55% after 6 weeks of storage, while these values were approximately of 70, 40, and 20% for mandarins treated with both F6/SB and 2% SB alone. Regarding disease severity, no statistical differences between treatments and control were found after 2 weeks of storage at 5 °C, while after 4 and 6 weeks, the treatments F10/SB, F6/SB, and 2% SB significantly and similarly reduced sour rot severity. Severity reductions obtained with the SB 2%, F10/SB, and F6/SB treatments were of 31, 48, and 22% and of 32, 53, and 43% after 4 and 6 weeks of storage, respectively.

### 3.2. Effect of Coating Application on Mandarin Quality during Cold Storage
3.2.1. Fruit Weight Loss

The weight loss percentages of uncoated (control), 2% aqueous SB-treated, and coated 'Orri' mandarins after 2, 4, and 6 weeks of cold storage at 5 °C, and also after the 1-week shelf-life periods at 20 °C following these storage periods, are shown in Figure 2. As expected, weight loss significantly increased during cold storage and shelf life for all treatments, although it was significantly lower on coated mandarins than on uncoated controls and SB-dipped fruits, which highlights the efficacy of these PPS-GMS coatings as moisture barriers. Weight loss was in the range of 4.1–5.7%, 6.0–8.0%, and 8.9–10.7% after 2, 4, and 6 weeks of cold storage followed by 1 week of shelf life, with uncoated controls and 2% SB-treated mandarins showing the highest values (10.7 and 9.8%, respectively, after 6 weeks at 5 °C plus 1 week at 20 °C). In general, during the whole storage period, the weight loss of mandarins coated with the F6 and F10 coatings was lower than that of those coated with the coatings F6/SB and F10/SB, although these differences were not always significant. This fact suggests that the addition of SB to the coating emulsions negatively influenced their ability to hold water vapor and maintain fruit weight. The F10 coating was the most effective treatment in reducing weight loss throughout the entire storage time, although in some cases it was not significantly superior to the F6 coating. In the

last evaluation, while weight loss of uncoated controls was of 10.7%, it was of 8.2% on F10-coated mandarins ($p < 0.05$).

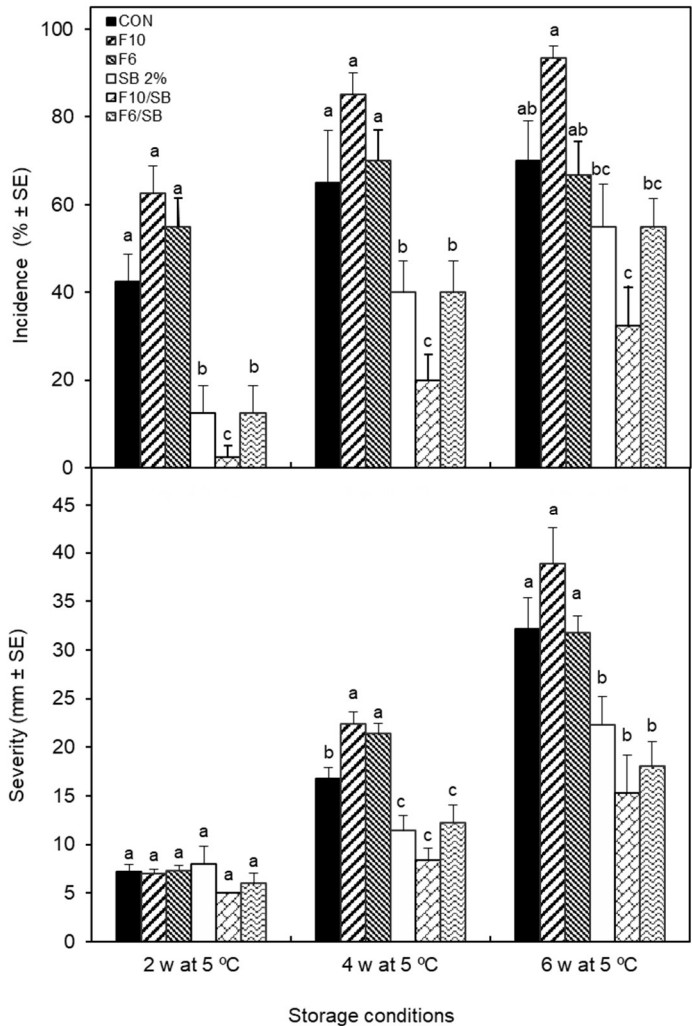

**Figure 1.** Incidence and severity of sour rot on 'Orri' mandarins artificially inoculated with *Geotrichum citri-aurantii*, dipped 24 h later in water (control, CON), 2% sodium benzoate aqueous solution (SB 2%), or coated with starch-monostearate edible composite coatings without (F10, F6) or with sodium benzoate (F10/SB, F6/SB). Inoculated and coated fruits were stored for up to 6 weeks at 5 °C and 90% RH. Values are means of 2 experiments, each with 4 replicates of 10 fruits per treatment. For each parameter and storage period, columns with different letters indicate significant differences according to Fisher's protected LSD test ($p < 0.05$) applied after an ANOVA. For disease incidence, the ANOVA was applied to the arcsine-transformed values. Non-transformed means are shown. w = weeks.

### 3.2.2. Fruit Firmness

As expected, values of fruit firmness as percentage rind deformation were higher after cold storage and shelf life than values at harvest (Table 2). Fruit firmness was found to be significantly influenced by the treatment applied during the entire storage period. After 2, 4, and 6 weeks of cold storage plus 1 week of shelf life, mandarins coated with the F10 and F6 coatings showed a lower percentage of deformation than uncoated control fruits, while fruits treated with 2% SB or coated with the F10/SB and F6/SB coatings in general did not statistically differ from controls. Overall, the firmness of mandarins treated with the F10 coating was significantly higher than that of the rest of treated fruits, especially after 2 and 4 weeks at 5 °C plus 1 week at 20 °C.

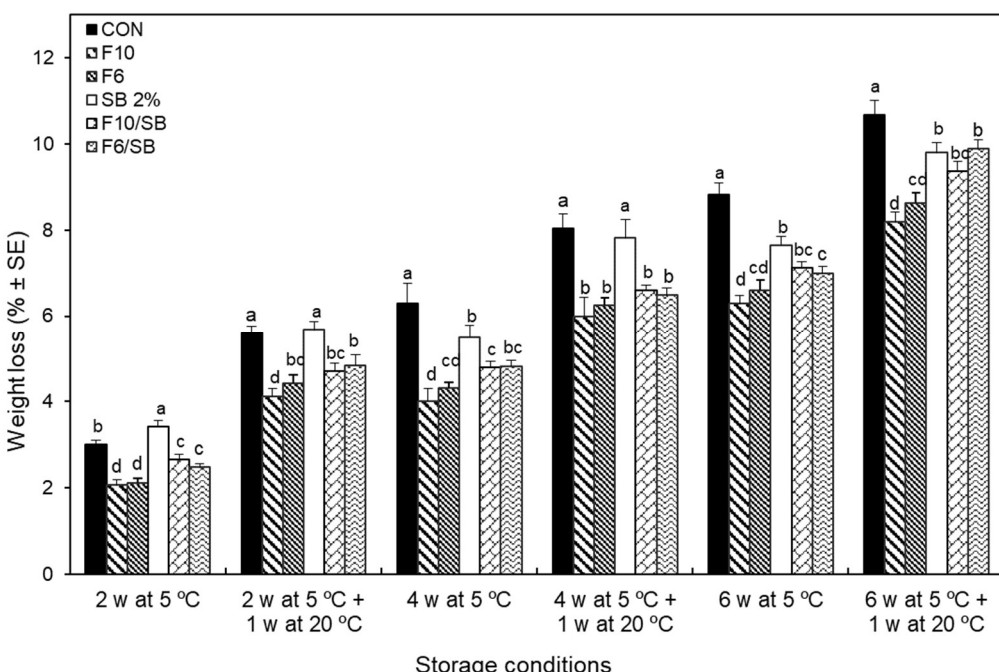

**Figure 2.** Weight loss of 'Orri' mandarins dipped in water (control, CON), 2% sodium benzoate aqueous solution (SB 2%), or coated with starch-monostearate edible composite coatings without (F10, F6) or with sodium benzoate (F10/SB, F6/SB), and stored at 5 °C followed by a shelf-life period of 7 days at 20 °C. For each storage period, columns with different letters are significantly different according to Fisher's protected LSD test ($p < 0.05$) applied after an ANOVA. w = weeks.

**Table 2.** Quality attributes of 'Orri' mandarins subjected to different postharvest treatments and stored for up to 6 weeks at 5 °C followed by 1 week of shelf life at 20 °C.

| Quality Attributes [2] | Storage Conditions [3] | Treatments [1] | | | | | |
|---|---|---|---|---|---|---|---|
| | | Control | F10 | F6 | SB 2% | F10/SB | F6/SB |
| Firmness (% deformation) | At harvest | 4.52 ± 0.62 | - | - | - | - | - |
| | 2 w at 5 °C + 1 w at 20 °C | 5.99 ± 0.83 [a] | 4.78 ± 0.50 [c] | 5.31 ± 0.75 [b] | 5.95 ± 0.92 [a] | 5.82 ± 0.79 [ab] | 5.63 ± 1.62 [ab] |
| | 4 w at 5 °C + 1 w at 20 °C | 7.34 ± 0.99 [a] | 5.45 ± 0.65 [c] | 5.89 ± 0.34 [bc] | 6.25 ± 0.91 [b] | 5.92 ± 1.14 [bc] | 6.21 ± 0.78 [b] |
| | 6 w at 5 °C + 1 w at 20 °C | 6.88 ± 0.85 [a] | 5.94 ± 0.80 [b] | 6.30 ± 0.42 [ab] | 6.44 ± 0.64 [ab] | 6.55 ± 0.81 [a] | 6.54 ± 0.39 [a] |
| TA (% citric acid) | At harvest | 0.54 ± 0.04 | - | - | - | - | - |
| | 2 w at 5 °C + 1 w at 20 °C | 0.46 ± 0.04 [b] | 0.51 ± 0.05 [ab] | 0.58 ± 0.05 [ab] | 0.50 ± 0.26 [ab] | 0.64 ± 0.07 [a] | 0.68 ± 0.14 [a] |
| | 4 w at 5 °C + 1 w at 20 °C | 0.45 ± 0.03 [b] | 0.41 ± 0.02 [b] | 0.50 ± 0.08 [ab] | 0.45 ± 0.05 [b] | 0.59 ± 0.05 [a] | 0.56 ± 0.05 [a] |
| | 6 w at 5 °C + 1 w at 20 °C | 0.57 ± 0.04 [a] | 0.54 ± 0.11 [ab] | 0.50 ± 0.07 [b] | 0.56 ± 0.04 [a] | 0.50 ± 0.02 [b] | 0.60 ± 0.06 [a] |
| SSC (°Brix) | At harvest | 11.58 ± 0.57 | - | - | - | - | - |
| | 2 w at 5 °C + 1 w at 20 °C | 11.77 ± 0.78 [b] | 12.88 ± 0.31 [a] | 12.91 ± 0.35 [a] | 11.82 ± 0.07 [b] | 13.18 ± 0.12 [a] | 13.37 ± 0.72 [a] |
| | 4 w at 5 °C + 1 w at 20 °C | 13.81 ± 0.63 [a] | 12.58 ± 0.23 [b] | 12.98 ± 0.76 [ab] | 13.46 ± 0.23 [a] | 12.55 ± 0.54 [b] | 12.37 ± 1.21 [b] |
| | 6 w at 5 °C + 1 w at 20 °C | 15.15 ± 0.11 [a] | 13.27 ± 0.11 [c] | 12.42 ± 0.32 [d] | 14.17 ± 0.23 [b] | 13.35 ± 0.21 [c] | 13.32 ± 0.22 [c] |
| AcC (mg/L) | At harvest | 5.6 ± 1.3 | - | - | - | - | - |
| | 2 w at 5 °C + 1 w at 20 °C | 12.3 ± 0.6 [c] | 14.7 ± 0.9 [bc] | 15.2 ± 2.8 [bc] | 12.8 ± 1.1 [c] | 19.4 ± 4.2 [a] | 15.9 ± 2.3 [b] |
| | 4 w at 5 °C + 1 w at 20 °C | 13.8 ± 1.3 [c] | 27.8 ± 2.2 [a] | 18.6 ± 3.2 [b] | 14.2 ± 2.0 [c] | 18.4 ± 1.3 [b] | 18.8 ± 2.1 [b] |
| | 6 w at 5 °C + 1 w at 20 °C | 15.1 ± 2.2 [c] | 22.4 ± 4.4 [ab] | 28.0 ± 6.4 [a] | 19.0 ± 1.7 [bc] | 18.1 ± 3.2 [bc] | 22.8 ± 7.1 [ab] |
| EtC (mg/L) | At harvest | 95.7 ± 21.3 | - | - | - | - | - |
| | 2 w at 5 °C + 1 w at 20 °C | 307.1 ± 35.4 [c] | 668.9 ± 177.7 [b] | 677.3 ± 57.8 [b] | 333.8 ± 11.6 [c] | 820.8 ± 216.5 [b] | 1013.1 ± 189.5 [a] |
| | 4 w at 5 °C + 1 w at 20 °C | 389.5 ± 17.8 [c] | 1451.0 ± 180.9 [a] | 1024.8 ± 245.1 [b] | 387.8 ± 70.5 [c] | 1369.9 ± 165.9 [a] | 1459.4 ± 62.9 [a] |
| | 6 w at 5 °C + 1 w at 20 °C | 756.4 ± 99.3 [b] | 1474.4 ± 253.3 [a] | 1403.1 ± 204.8 [a] | 733.5 ± 127.0 [b] | 1438.7 ± 248.5 [a] | 1589.3 ± 418.8 [a] |

[1] Control: uncoated, dipped in water; SB 2%: dipped in 2% sodium benzoate aqueous solution; F10, F6: coatings formulated without sodium benzoate; F10/SB, F6/SB: coatings formulated with sodium benzoate. [2] TA: titratable acidity; SSC: soluble solids content, EtC: ethanol content, AcC: acetaldehyde content. [3] w = weeks. Means (± SD) in rows with different superscript letters are significantly different according to Fisher's protected LSD test ($p < 0.05$) applied after an ANOVA.

### 3.2.3. Quality of Mandarin Juice

Mandarin juice quality during storage was determined in terms of TA, SSC, AcC, and EtC (Table 2). After 2 and 4 weeks at 5 °C plus 1 week at 20 °C, mandarins coated with

F10/SB and F6/SB showed significantly higher TA (0.59 and 0.56% citric acid, respectively, after 4 weeks) than uncoated samples (0.45%), but the difference was not significant after 6 weeks. The application of coatings containing SB (F10/SB and F6/SB) generally induced higher TA in the mandarin juice than the rest of treatments, including dipping in 2% aqueous SB. Overall, irrespective of the postharvest treatment, SSC in mandarin juice increased during the storage period from the value at harvest (11.58 °Brix). While coated fruits showed higher SSC than uncoated controls and 2% SB-treated fruits after 2 weeks at 5 °C plus 1 week at 20 °C, the opposite was observed after 4 and 6 weeks of cold storage followed by 1 week of shelf life. For example, values of about 15 and 13 °Brix for uncoated and coated mandarins, respectively, were obtained after 6 weeks. No significant differences in SSC were observed among different coating formulations in most of the evaluation dates. The mandarin juice contents of ethanol and acetaldehyde increased from the initial 5.6 and 95.7 mg/L (values at harvest) to 15.1–28.0 and 756.4–1589.3 mg/L, respectively, at the end of the entire cold storage period and shelf life. The use of PPS-GME-based coatings, both with and without SB, significantly increased the content of both volatiles in the juice of coated 'Orri' mandarins ($p < 0.05$) compared to uncoated controls and mandarins treated with 2% SB aqueous solutions. At the end of the whole storage period, after 6 weeks of cold storage at 5 °C followed by 1 week at 20 °C, no significant differences in both AcC and EtC were found among fruits subjected to coating treatments.

### 3.2.4. Internal Gas Concentration

The concentrations of internal $CO_2$ and $O_2$ within control (uncoated), 2% SB-treated, and coated mandarins during cold storage plus shelf life are presented in Figure 3. After each evaluation period, every tested coating modified the internal atmosphere of 'Orri' mandarins, inducing an increase in $CO_2$ and a decrease in $O_2$ in comparison to those values in uncoated fruits. In general, the addition of SB into the emulsions F10 and F6 did not strongly modify the effect of the coating on the internal gas concentration of coated mandarins. Similarly, no marked differences in gas levels were observed between fruits treated with the coatings F10 and F6 and F10/SB and F6/SB. The contents of internal $CO_2$ and $O_2$ in coated mandarins during storage reached values of approximately 3–5 and 11–19 kPa, respectively.

### 3.2.5. Sensory Analysis

Sensory analysis showed that after 2 weeks of storage at 5 °C plus 1 week at 20 °C of shelf life, the overall flavor of coated mandarins was rated as acceptable, with scores ranging from approximately 5.2–5.7 (Table 3). At this evaluation date, there were no significant differences among treatments, with a score of 5.6 for uncoated control fruits. However, after 4 and 6 weeks and the corresponding shelf-life periods, the judges detected significant differences in overall flavor among control fruits (5.6–5.8) and F10/SB and F6/SB-treated mandarins (4.0–4.5), although the scores of the latter still indicated an acceptable flavor. Irrespective of the storage period, the flavor of mandarins dipped in 2% SB solution was similar to that of control fruits, with scores in the range of 5.2–5.7. With respect to the presence of off-flavors, the judges only detected slight off-flavor (scores in the range 2.1–2.5) in mandarins coated with the F10/SB and F6/SB coatings after 4 and 6 weeks of storage at 5 °C plus 1 week at 20 °C. Fruits subjected to the rest of treatments always scored below 2.0.

Regarding the external appearance of treated mandarins, no differences were observed among treatments, including uncoated control fruits, during the whole storage period (Table 3). All samples were scored as fair–good and ranged from 2.6 to 1.9. Nevertheless, significant differences in fruit gloss were found among treatments (Table 4). After 2, 4, and 6 weeks of cold storage plus 1 week of shelf life, F6/SB-coated samples were ranked with the lowest gloss, while F10-coated fruits were ranked as the glossiest. After 6 weeks plus shelf life, only mandarins coated with the F10/SB and F6/SB coatings significantly showed lower gloss than mandarins subjected to the rest of treatments, including uncoated controls.

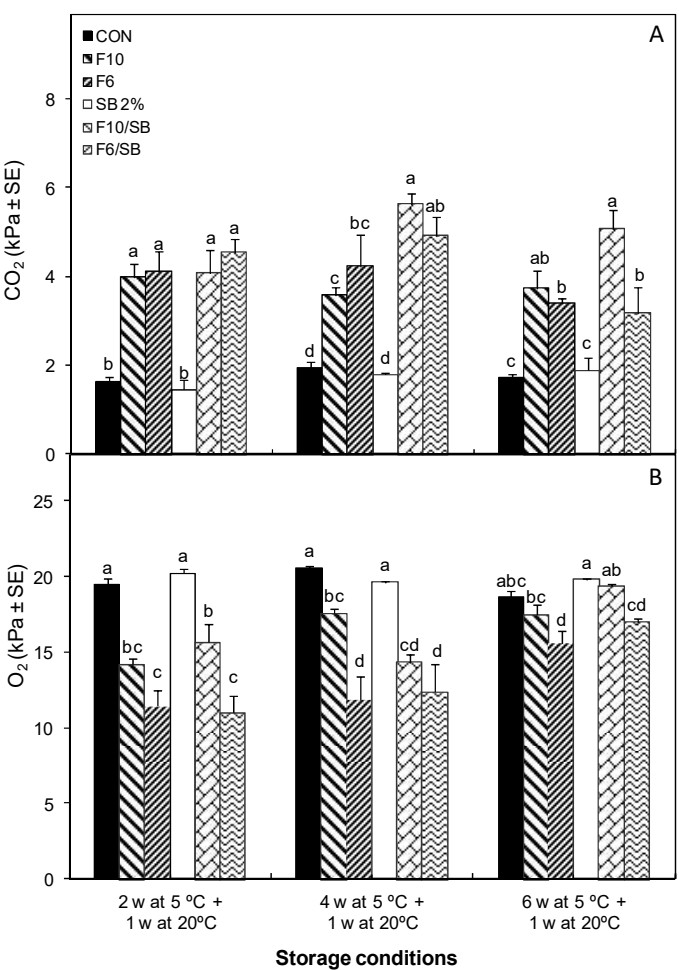

**Figure 3.** Internal $CO_2$ (**A**) and $O_2$ (**B**) concentrations in 'Orri' mandarins dipped in water (control, CON), 2% sodium benzoate aqueous solution (SB 2%), or coated with starch-monostearate edible composite coatings without (F10, F6) or with sodium benzoate (F10/SB, F6/SB), and stored at 5 °C followed by a shelf-life period of 7 days at 20 °C. For each parameter and storage period, columns with different letters are significantly different according to Fisher's protected LSD test ($p < 0.05$) applied after an ANOVA. w = weeks.

**Table 3.** Sensory attributes of 'Orri' mandarins subjected to different postharvest treatments and stored for up to 6 weeks at 5 °C followed by 1 week of shelf life at 20 °C.

| Sensory Attributes | Storage Conditions [2] | Treatments [1] | | | | | |
|---|---|---|---|---|---|---|---|
| | | Control | F10 | F6 | SB 2% | F10/SB | F6/SB |
| Overall Flavor [3] (1–9 scale) | At harvest | 6.5 ± 1.38 | - | - | - | - | - |
| | 2 w at 5 °C + 1 w at 20 °C | 5.6 ± 1.65 a | 5.5 ± 1.81 a | 5.7 ± 1.81 a | 5.7 ± 1.81 a | 5.2 ± 1.47 a | 5.7 ± 1.55 a |
| | 4 w at 5 °C + 1 w at 20 °C | 5.8 ± 1.37 a | 4.7 ± 1.40 b | 5.5 ± 1.25 ab | 5.2 ± 1.26 ab | 4.5 ± 1.41 b | 4.5 ± 1.78 b |
| | 6 w at 5 °C + 1 w at 20 °C | 5.6 ± 1.76 a | 4.4 ± 1.59 ab | 4.9 ± 1.62 ab | 5.6 ± 1.72 a | 4.3 ± 2.09 b | 4.0 ± 1.65 b |
| Off-Flavor [4] (1–5 scale) | At harvest | 1.1 ± 0.34 | - | - | - | - | - |
| | 2 w at 5 °C + 1 w at 20 °C | 1.5 ± 0.74 a | 1.6 ± 0.98 a | 1.7 ± 0.80 a | 1.5 ± 0.64 a | 1.6 ± 0.84 a | 1.5 ± 0.84 a |
| | 4 w at 5 °C + 1 w at 20 °C | 1.4 ± 0.74 b | 1.9 ± 0.96 ab | 1.4 ± 0.83 b | 1.6 ± 0.91 b | 2.5 ± 1.13 a | 2.1 ± 1.25 ab |
| | 6 w at 5 °C + 1 w at 20 °C | 1.3 ± 0.62 c | 1.7 ± 1.26 bc | 1.3 ± 1.06 c | 1.5 ± 0.74 c | 2.3 ± 1.19 ab | 2.5 ± 1.13 a |
| External Aspect [5] (1–3 scale) | At harvest | 2.5 ± 0.67 | - | - | - | - | - |
| | 2 w at 5 °C + 1 w at 20 °C | 2.4 ± 0.85 a | 2.5 ± 0.76 a | 1.9 ± 0.83 a | 2.4 ± 0.76 a | 2.1 ± 0.77 a | 2.4 ± 0.74 a |
| | 4 w at 5 °C + 1 w at 20 °C | 2.4 ± 0.63 a | 2.6 ± 0.63 a | 2.4 ± 0.65 a | 2.4 ± 0.74 a | 2.6 ± 0.65 a | 2.3 ± 0.83 a |
| | 6 w at 5 °C + 1 w at 20 °C | 2.5 ± 0.52 a | 2.3 ± 0.61 a | 2.4 ± 0.65 a | 2.4 ± 0.65 a | 2.1 ± 0.78 a | 2.2 ± 0.80 a |

[1] Control: uncoated, dipped in water; SB 2%: dipped in 2% sodium benzoate aqueous solution; F10, F6: coatings formulated without sodium benzoate; F10/SB, F6/SB: coatings formulated with sodium benzoate. [2] w = weeks. [3] Overall flavor scale: 1–3 = bad, 4–6 = acceptable, 7–9 = excellent. [4] Off-flavors ranked from 1 (absence) to 5 (presence). [5] Coating/fruit appearance ranked from 1 (bad) to 3 (good). Means (±SD) in rows with different letters are significantly different according to Fisher's protected LSD test ($p < 0.05$) applied after an ANOVA.

**Table 4.** Ranked fruit gloss of 'Orri' mandarins subjected to different postharvest treatments [1] and stored for up to 6 weeks at 5 °C followed by 1 week of shelf life at 20 °C.

| Ranked Fruit Gloss | Storage Conditions [2] | | | | | |
|---|---|---|---|---|---|---|
| | 2 w at 5 °C + 1 w at 20 °C | | 4 w at 5 °C + 1 w at 20 °C | | 6 w at 5 °C + 1 w at 20 °C | |
| More glossy | F10 | a | F10 | a | F10 | a |
| | F6 | a | F6 | a | Control | a |
| | Control | a | Control | a | F6 | ab |
| | SB 2% | b | SB 2% | b | SB 2% | b |
| | F10/SB | b | F10/SB | bc | F10/SB | c |
| Less glossy | F6/SB | c | F6/SB | c | F6/SB | c |

[1] Control: uncoated, dipped in water; SB 2%: dipped in 2% sodium benzoate aqueous solution; F10, F6: coatings formulated without sodium benzoate; F10/SB, F6/SB: coatings formulated with sodium benzoate. [2] w = weeks. Treatments in columns with different letters are significantly different according to Friedman test ($p < 0.05$) ($n = 15$).

## 4. Discussion

This research highlights the ability of antifungal PPS-GSM edible coatings formulated with 2% SB to control postharvest sour rot on artificially inoculated mandarin fruits during cold storage. The results showed that the coatings formulated without SB not only did not exhibit ability to reduce sour rot, but rather favored the development of *G. citri-aurantii* on coated mandarins. This pattern was more evident with the F10 than with the F6 coating. These composite coating matrixes contain the same ingredients but in different proportions. Specifically, main ingredients PPS, GMS, and glycerol are 1, 1, and 0.5% (*w/w*) in F6 whereas they are 2, 0.5, and 1% in F10 [21]. Therefore, F10 contains two times more PPS than F6, which can provide an explanation for the obtained results. It is known that several species of filamentous fungi can break down starch into monosaccharides through the action of amylase enzymes [23]. For instance, *G. candidum*, a fungal species very close to *G. citri-aurantii*, has the ability to produce and secrete large amounts of amylases, and can break down starch into glucose [24]. Glucose can be further degraded into low-carbon sugars and energy through the glycolysis pathway. Hence, we suggest that the starch present in the F6 and particularly in the F10 coatings might serve as a carbohydrate source for *G. citri-aurantii*, which caused a higher disease incidence and severity on coated mandarins than on uncoated controls. By contrast, the same coatings formulated with 2% SB significantly controlled sour rot on mandarins, which indicates that the incorporation of the SB salt into the coating matrix was responsible for the antifungal activity of the coatings F10/SB and F6/SB. This result confirms the ability of the GRAS salt SB to reduce disease caused by *G. citri-aurantii* on cold-stored citrus fruits, as reported in previous studies with mandarins [21] and lemons [25] incubated at ambient temperatures. Moreover, it was also found in previous work that these matrix coatings formulated with SB were able to reduce other important citrus postharvest diseases such as green and blue molds, caused by *P. digitatum* and *P. italicum*, respectively, on mandarins, oranges, and lemons either incubated at room temperature or stored at low temperatures [26]. On the other hand, in the present work, the incorporation of SB within the F10 emulsion (to form the coating F10/SB) improved the antifungal effectiveness when compared to the application of SB as an aqueous solution. Mehyar et al. [20] similarly reported that 0.1% potassium sorbate (PS, another GRAS salt) incorporated into pea starch and guar-gum-based coatings was more effective in inhibiting spoilage molds on cold-stored cucumbers, tomatoes, and apples than a PS aqueous solution. Interestingly, F10/SB was remarkably more effective than F6/SB for sour rot control on mandarins. A key factor that may strongly influence this performance is the difference of viscosity between the two emulsion formulations. F10/SB has higher viscosity (33.6 cP) than F6/SB (11.2 cP), which might result in a larger effective load of SB on the surface of treated mandarins via the formation of a thicker coating layer [26]. However, more studies are necessary to determine how the matrix composition influence the availability of SB in PPS-based coatings, its release ability, and its diffusion into the

fruit rind wounds that are the fungal infections courts. It is interesting to note that the addition of SB to the coating matrixes considerably reduced the viscosity of the resulting emulsion, which was originally 55.1 and 48.6 cP for the coatings F10 and F6, respectively. Even though we did not specifically study the causes of this reduction in viscosity, we consider that the interaction of the organic salt SB with PPS was the most important factor affecting this change, because, in general, the viscosity of composite coatings is majorly given by the amount and properties of the hydrocolloid component of the matrix, PPS in this case, although the lipid ratio and the type of lipid and other minor components can also play a role [13].

The impact of the application of antifungal coating emulsions on the physicochemical and sensorial quality of 'Orri' mandarins cold-stored at commercial refrigeration temperatures for relative long periods followed by the simulation of commercial shelf-life periods (1 week at 20 °C) was the second general objective of this research work. In terms of economical return to citrus growers, the maintenance of overall fruit quality, and particularly, the reduction in fruit weight loss, during the whole period between harvest and final consumption is a key factor that fully justifies the commercial implementation of postharvest treatments such as waxes or ECs. In this study, the best reduction in weight loss was achieved on the mandarins treated with the F10 coating, and as mentioned above, the F10 coating contains more PPS and less GMS than the F6 coating. This result confirms those obtained in a previous study of optimization of the formulations used in this study, in which it was found that an increase in PPS concentration in the EC formulation together with the use of low GMS content (0.5%) reduced mandarin weight loss after 2 weeks of storage at 20 °C [21]. Thus, the present results corroborate the efficacy of this coating to reduce fruit weight loss, also on citrus fruits subjected to long periods of cold storage followed by a shelf-life period at room temperatures. In general, starch-based films are considered to be water sensitive and exhibit only medium or low water vapor barrier capacity, which is the reason why the incorporation of other hydrophobic or lipidic compounds as additional coating ingredients are highly recommended to contribute to increase the water vapor barrier properties and the water resistance of the resulting films [15,27]. On the other hand, the results showed that the addition of SB to the F10 and F6 coatings to form the F10/SB and F6/SB formulations was detrimental to the capacity of the coatings to reduce weight loss on coated mandarins, as can be observed in Figure 2. Thus, the water barrier properties of the coatings were considerably influenced by the addition of the salt SB at 2% to the PPS-GMS matrix. In this sense, prior research works showed that the incorporation of GRAS salts, such as SB, PS, and bicarbonates or carbonates, to EC emulsion matrixes of different nature considerably influenced the moisture barrier properties of coating films in vitro and also of ECs applied to fresh horticultural products, such as citrus, persimmons, and cherry tomatoes [28–30]. Therefore, it is important to consider the specific interactions of additional ingredients such as GRAS salts or food additives with main coating matrix ingredients, as they might change some intrinsic emulsion properties, such as viscosity and pH, and affect the interfacial behavior and the wettability of the coating emulsions [29,31]. These interactions in conjunction to intrinsic fruit characteristics will eventually determine the real impact of coating application to weight loss and other external fruit quality attributes such as firmness [13,27].

In general, the firmness of 'Orri' mandarins was better maintained during cold storage and shelf life on coated than uncoated (control) mandarins. However, as it was also observed for weight loss, the PPS-GMS ECs formulated without SB, and particularly F10, were superior for fruit firmness maintenance to the ECs formulated with SB. This result was not unexpected since it was reported that the effect of the application of EC emulsions on the firmness of citrus fruits after harvest is typically related to their effect on fruit weight loss [32]. For instance, Navarro-Tarazaga et al. [33] observed a direct relationship between fruit firmness and weight loss on 'Ortanique' mandarins coated with hydroxypropyl methylcellulose (HPMC)-beeswax (BW) and oleic acid and stored at low temperature. In the present work, the coating F10 (without SB) was superior for firmness

and weight loss control on mandarins stored at 5 °C compared to the rest of the treatments. Nonetheless, the impact of ECs application on fruit firmness and weight loss depends not only on the coating composition and properties but also on the inherent characteristics of the citrus species, and even cultivar, and the postharvest storage conditions. Thus, in another study by our group, the effect of the application of this kind of PPS-GMS-based coating amended with SB on firmness and weight loss of 'Fino' lemons during storage for up to 4 weeks at 12 °C (adequate temperature for commercial storage of lemons) plus 1 week of shelf life at 20 °C was not significantly different than the effect of the application of the coatings without SB [25], and this can only be attributed to differences in the storage temperature (12 °C vs. 5 °C) and the characteristics of the fruit rind (lemons vs. mandarins). Further evidence of the influence of the type of coating, but also of the type of fresh citrus fruits and the storage temperature, are given in previous recent research conducted on oranges and other mandarin cultivars with ECs formulated with other coating matrixes, such as HPMC-BW and pectin-BW, and other antifungal ingredients, such as some selected essential oils [34,35].

Regarding mandarin juice quality, the values of TA and SSC during the whole storage and shelf-life periods followed a similar pattern for all coated mandarins regardless of coating type. It is clear from the data (Table 2) that coating considerably contributed to the maintenance of the TA measured at harvest, if compared to that of uncoated controls, and that this effect was not dependent on the addition of SB to the coating matrixes. At the first evaluations, after 2 and 4 weeks of storage plus shelf life, TA levels in coated mandarins were significantly higher than in mandarins dipped in water (controls) or in 2% SB solution, while after the last evaluation, after 6 weeks plus shelf life, these differences in TA were not observed. At this time, SSC was considerably lower in coated than in uncoated mandarins, indicating a general delay of the fruit ripening processes in coated fruits compared to uncoated samples. Therefore, in the case of 'Orri' mandarins stored at 5 °C, the tested PPS-GMS coatings influenced the fruit respiration rate, the organic acid profiles, and the conversion of starch and acids into sugars throughout the storage period [36], most likely through the capacity of starch-based films to create a semi-permeable gas barrier around the fruit [15,27]. Similar findings were observed with cold-stored 'Valencia' oranges treated with an EC based on guar gum and pea starch [37]. Again, however, the influence of coating on fruit ripening during storage is not only dependent on the coating itself but also on the fruit species and cultivar and the storage conditions. Thus, in contrast to the present findings with 'Orri' mandarins, in our previous research work with 'Fino' lemons stored at 12 °C for up to 4 weeks plus 1 week of shelf life, no significant differences in TA and SSC were found between uncoated lemons and lemons coated with PPS-GMS-based emulsions, irrespective of the presence of SB as an antifungal ingredient [25].

Among citrus fruits, mandarins are generally very prone to the development of off-flavors during prolonged storage due to their high respiration rate. Therefore, mandarins can often serve as a sensitive indicator for the gas exchange capacity of ECs applied as postharvest treatments. Fruit coating can have the capacity to extend the fruit postharvest life by regulating fruit respiration and modifying the levels of $O_2$ and $CO_2$ within the fruit (internal gas concentration) [13]. Nevertheless, if the internal $O_2$ concentration becomes too low, fermentation processes can occur, resulting in abnormally high values of the content of the volatiles ethanol and acetaldehyde in the juice, which can lead to the production of off-flavors [32]. In this study, the production of both ethanol and acetaldehyde in mandarins increased after the application of all PPS-GMS coatings compared to the production of these volatiles in uncoated fruits (both controls dipped in water and fruits dipped in SB aqueous solution), which confirms the effective creation of a modified atmosphere within coated fruits. However, the maximum value of EtC measured analytically in coated mandarins in these experiments was 1590 mg/L after cold storage for 6 weeks plus 1 week of shelf life, corresponding to the F6/SB-treated fruits, and this value is considerably lower than the EtC typically associated with off-flavor production in citrus fruits, i.e., 2000 mg/L [38]. In addition, only very slight off-flavors were detected in coated mandarin samples by the

judges in the sensory analysis, and the overall flavor was ranked as acceptable for all coating treatments. Overall, it was found in the physicochemical and sensory evaluations that the addition of SB to the PPS-GMS-based coatings slightly modified the gas exchange properties of the films, affecting to a greater extent the internal gas concentration and increasing the volatile content in the juice, but the impact of such modifications was not important enough to affect the overall flavor and induce important off-flavors in the mandarins coated with these films containing SB. In this sense, similar results were obtained with coating matrixes of different natures to which SB or other GRAS salts were incorporated as antifungal ingredients and were applied as postharvest treatments to citrus fruits, subjected later to long-term cold storage [17,25,34]. On the other hand, although in some evaluations the differences were not significant, these adverse modifications were generally more important for the coating F6 than for the coating F10. Although the external visual appearance of the mandarins was rated as fair–good for all treatments, it was observed that the coating F10 provided the best gloss to coated mandarins, but the addition of SB to the emulsions negatively affected the optical properties of the coatings, reducing their capacity to give gloss to coated fruit, especially in the case of the F6 coating.

## 5. Conclusions

Several PPS-GMS composite coatings, formulated with or without SB as an antifungal ingredient, were evaluated in this research to control citrus sour rot and maintain the quality of 'Orri' mandarins cold-stored at 5 °C for up to 6 weeks followed by a shelf-life period of 1 week at 20 °C. The PPS-GMS formulation amended with SB and encoded as F10/SB was the most promising EC to reduce the incidence and severity of sour rot caused by *G. citri-aurantii* on artificially inoculated mandarins and could represent a feasible replacement to the use of polluting synthetic fungicides, such as guazatine and propiconazole, and citrus conventional waxes. Overall, the F10/SB coating showed the greatest potential for commercial use as an efficient and environmentally friendly alternative as it also reduced weight loss in mandarins with no negative effects on the physicochemical and sensory quality of treated fruits. The composition of the F10/SB coating is as follows (d.b.): 32% PPS, 8.8% GMS, 16.7% glycerol, and 2% SB (w.b.), with a total solids content of 5.9%.

**Author Contributions:** Conceptualization: L.P. and M.B.P.-G.; methodology: L.P. and M.B.P.-G.; validation: V.M.-B., L.S.-M. and M.B.P.-G.; formal analysis: L.S.-M., V.M.-B. and M.B.P.-G.; investigation: L.S.-M., V.M.-B., M.B.P.-G. and L.P.; resources: L.P. and M.B.P.-G.; data curation: L.S-M., V.M.-B., M.B.P.-G. and L.P.; writing—original draft preparation: L.S.-M.; writing—review and editing: L.S.-M., M.B.P.-G. and L.P.; visualization: L.S.-M.; supervision: L.P. and M.B.P.-G.; project administration: L.P. and M.B.P.-G.; funding acquisition: L.P. and M.B.P.-G. All authors have read and agreed to the published version of the manuscript.

**Funding:** This work was partially funded by the StopMedWaste project (EU PRIMA Programme-2019; NextGenerationEU/PRTR; Spanish "Agencia Estatal de Investigación", PCI2020-112095). Additional funding was received from the IVIA (Project No. 52201) and the EU European Regional Development Fund (ERDF) of the Generalitat Valenciana 2021–2027. Lourdes Soto-Muñoz's postdoctoral program was supported by a scholarship by the Mexican National Council of Science and Technology (CONACYT-160058-México). Victoria Martínez-Blay's research scholarship was supported by the IVIA and the European Social Fund ('Beca IVIA-FSE' 2018 No. 24).

**Institutional Review Board Statement:** Not applicable.

**Informed Consent Statement:** Not applicable.

**Data Availability Statement:** The data presented in this study belong to the IVIA and are available on request from the corresponding author.

**Acknowledgments:** Fontestad S.A. (Montcada, Valencia, Spain) is gratefully acknowledged for providing fruit and technical assistance.

**Conflicts of Interest:** The authors declare no conflict of interest.

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
