# Peer review of "Postharvest Application of Potato Starch Edible Coatings with Sodium Benzoate to Reduce Sour Rot and Preserve Mandarin Fruit Quality"

_coatings, doi:10.3390/coatings13020296_

Round 1
Reviewer 1 Report
1. The abstract must be modified. The abstract must be rearranged and contained these elements (Abstract Purpose Design/ methodology/ approach Findings Originality/value) for this article. Please rearrange the introduction to clear your aim and the novelty.
2. The analysis of Fruit Firmness results in the manuscript should be elaborated more clearly and accurately by citing references.
3. There are some errors in the format of temperature units in the manuscript, please check and correct.
4. Line 94-97, " Nevertheless, when these coating matrixes are formulated with additional functional ingredients, like antimicrobial or antioxidant agents, the addition of new compounds might lead to unstable or incompatible emulsions showing phase separation, modify the functional properties of the coating, and/or change the or-ganoleptic profile of the coated product." Author should reframe the sentence to make it clearer to understand. And give the suitable reference for the same.
5. The English language error throughout the manuscript should be corrected.
Reviewer 2 Report
The submitted manuscript entitled “Postharvest Application of Potato Starch Edible Coatings with Sodium Benzoate to Reduce Sour Rot and Preserve Mandarin Fruit Quality” is dealing with testing the performance of developed edible coatings based on potato starch and glyceryl monostearate. The study complements the previous authors’ works on this topic. The text has a reasonable organization supported with sufficient number of relevant references.
After consideration the below mentioned minor issues, I suggest the acceptation of given manuscript in Coatings Journal.
· If the formulations of tested edible coatings are completely the same as in the previous studies (doi: 10.1002/jsfa.11414; doi: 10.36253/phyto-12528), it should be mentioned and referred in the appropriate section 2.2.
· Content of sunflower lecithin and mono-diglyceride esters could be included in Table 1.
· Line 110, 498: in vitro should be written in italics.
Reviewer 3 Report
In this manuscript of “Postharvest Application of Potato Starch Edible Coatings with Sodium Benzoate to Reduce Sour Rot and Preserve Mandarin Fruit Quality”, the author evaluated the potential of two edible coating emulsions based on pregelatinized potato starch (PPS) and glyceryl monostearate (GMS) alone (F6 and F10) or formulated with the food additive sodium benzoate (SB, 2%) (F6/SB and F10/SB) to control sour rot, an important citrus postharvest disease caused by the fungus Geotrichum citri-aurantii, and maintain postharvest quality of cold-stored ‘Orri’ mandarins. The result showed that the greatest potential for commercial use as an efficient and environmentally friendly alternative to conventional fungicides and waxes for sour rot control and quality preservation of cold-stored mandarins. The research is meaningful and the manuscript is perfect,but i have some questions about the manuscript. The following are the comments:
1. In the introduction part, i think the author had better explain the meaning of F10 and F6 briefly.
2. Can the author tell us how to determine the proportion of the components of “Table 1”?
3. In fig.1, severity of sour rot on “Orri” mandarins, why the positive group is higher than the negative group? The figure can add (A) and (B) in it.
4. Please explain the advantage of F10/SB compared to F10 in a few words.
Reviewer 4 Report
The manuscript entitled "Postharvest Application of Potato Starch Edible Coatings with Sodium Benzoate to Reduce Sour Rot and Preserve Mandarin Fruit Quality", showed interesting results about the best way to protect the Mandarin from sour rot.
1. Line 23 and 24 check the °C should be without line under the degree and change it in all the manuscript.
2. Line 168 the p.d is mentioned in the table legend but inside the table no? Please check it.
4. How many days the obtained emulsions stay stable without separation?
5. Line 470-472 Please add more explanation about why by add SB into coating solution the emulsion viscosity was decreased significantly? I suggest to add the Viscosity results at Table 1 even if it from your previous work Ref: 26.
6. Line 356 "after the second and third evaluation periods" I suggest to rewrite the phrase mention the week.
7. I suggest to add the Mandarin pictures at zero day and after 2 weeks than 6 weeks of storage by comparing the uncoated with the coated one.
Regards,
Round 2
Reviewer 1 Report
accept